# Multilateral Debt Relief for Clean Ocean Energy

Anastasia Telesetsky

Department of Natural Resources and Environmental Sciences, California State Polytechnic University San Luis Obispo, San Luis Obispo, CA 93407, USA; ateleset@calpoly.edu

**Abstract:** As states bring more and more offshore wind online and build renewable energy capacity, the promise of large-scale ocean renewables such as offshore wind is not shared equally across all coastal states. This paper examines the situation of coastal states identified by the World Bank as Heavily Indebted Poor Countries (HIPCs) in the context of the boom in offshore wind investment. Specifically, the paper looks at the limited access to renewable energy production exacerbated by ongoing public debt loads, and the almost complete lack of access to clean ocean energy development for the poorest coastal states. Using statistics from the International Renewable Energy Agency and datasets from the Our World in Data project, this paper highlights that the most indebted coastal states only have access to 0.69% of the available renewable energy even though these states represent 4.6% of the global population. In the context of state responsibility for failing to meet climate obligations under the UNFCCC, this paper argues that a sovereign debt relief package offers an equitable remedy to HIPC coastal states, many of whom owe a substantial portion of their GDP as external public debt. The debt service payments would be invested in the upfront capital costs of ocean-based clean energy. These types of debt relief arrangements address international state responsibility and offer the dual co-benefits of long-term economic development and low-carbon sustainability.

**Keywords:** ocean renewable energy; coastal states; debt relief; state responsibility; climate debt; sovereign debt; renewable energy transition; economic sustainability; environmental sustainability; heavily indebted poor countries

## 1. Introduction

We are in the midst of a global energy transition to large-scale renewables. Of particular excitement for many coastal states is the growth in offshore wind deployment. According to the International Renewable Energy Agency (IRENA), large-scale wind farms will increase from 34 gigawatts (GW) of installed capacity in 2019 to 382 GW by 2030 and 2002 GW by 2050 to keep global warming below 1.5 degrees Celsius [1].

Offshore wind offers a significant source of reliable energy with turbines generating more power than a comparable onshore farm. By 2035, the capacity for an offshore wind turbine is anticipated to be as large as 17 MW of production per turbine in contrast to the capacity for a single onshore wind turbine of 5.5 MW [2]. Offshore wind development offers other advantages over onshore wind development. Offshore wind is also usually strongest in the afternoon and evening when consumer demand is at a peak in contrast to land-based wind resources that are most available at night. With concentrated population centers along the coast of many states, offshore wind development may be able to shorten transmission lines to places where there is a high peak demand, thereby simplifying some aspects of power delivery infrastructure.

Governments and energy corporations are making substantial investments globally in offshore wind with offshore wind accounting for 12% of renewable energy investments in 2020 (USD 41 billion), 9% in 2021 (USD 39 billion), and 7% in 2022 (USD 34 billion) [3]. Yet, many coastal states are not able to participate in this "energy bonanza" because they do not have the financial resources to invest in large-scale energy infrastructure which has high upfront capital costs [4]. Many states, particularly some of the poorest states, are

heavily indebted to international creditors. Because a number of these states also have high climate vulnerability, researchers have recently called for USD 520 billion in debt relief to help states meet climate adaptation goals and to "frontload" energy transition investments for indebted nations [5]. Researchers urge decision makers to address "climate debt" by developing new frameworks for managing debt including "programs for a sovereign debt restructuring mechanism and debt-for-climate swaps" [6] (p. 100).

States have tried to be creative with their financing. For example, Benin, which is categorized as a Heavily Indebted Poor Country (HIPC) by the World Bank, has issued sustainable development goal bonds to work towards development objectives including low-carbon electricity and zero-emission transportation [7]. These bonds do not, however, address sovereign debt loads.

Other non-conventional programs have been proposed to address sovereign indebtedness. Debt-for-conservation programs have been piloted [8–10] as well as debt-for-development (health/education) swaps [11]. Scholars and researchers have encouraged the development of new instruments particularly to allow an exchange of sovereign debt for climate projects [12–16]. Researchers have observed that debt relief for climate programs will need to be substantially large to make an economic difference for states or otherwise the relief will not provide the needed government budget to deliver effective conservation programs [15].

No large-scale "debt relief for energy" programs have been implemented though researchers and analysts have proposed using existing debt swap models to support other development outcomes. A U.S. Agency of International Development report on climate financing observed that "debt-for-nature swaps" can be "also used to finance renewable energy development; utility-scale battery storage, compensation for early shutdown of coal-fired power plants; energy efficiency improvements in the industry" and other sustainable development goals [16] (p. 110).

Focusing on the situation for low-income and lower-middle-income states, this article highlights the concept of "climate debt" as part of state responsibility for failing to mitigate climate emissions and argues that it is incumbent on international public creditors to create large-scale debt relief for clean energy programs to allow for both equitable development and future energy sustainability. This article will focus in particular on the potential of offshore wind sources of long-term clean energy for low-income coastal states with high wind energy capacity. This paper proceeds with a discussion of the datasets consulted, observations about the sustainable energy gap (e.g., 4.6% of the global population living in HIPCs has access to 0.69% of available renewable energy), discussion of sovereign debt creditor state responsibility for failure to mitigate climate emissions, and a proposal for debt relief in exchange for ocean-based renewable energy investments for states where offshore wind is a viable source of energy.

## 2. Methods

This study relied on publicly available datasets to explore the relationship between offshore wind capability, energy access, cumulative carbon emissions, and financial debt. The datasets included reports from the International Renewable Energy Agency (IRENA) and data collected by Our World in Data based on a variety of sources including United Nations agencies and the International Energy Agency. Each of the tables in the discussion below includes citations to the datasets. The IRENA data on offshore wind are from the 2023 Renewable Energy Statistics which used 2021 production data and 2022 capacity data. While it would be preferable to have data from one year, the data relied upon for this study still demonstrate general national trends in offshore production and capacity for most nations. The most recent data from the Our World in Data tables on per capita energy/electricity use, per capita emissions, and national renewable electricity production are from 2021. The data on national debt and GDP are from the United Nations Conference on Trade and Development from 2010 to 2022. Source information is available in the Data Availability Section.

## 3. Discussion

This section starts with a discussion of offshore wind and the inability of most low and lower-middle-income coastal countries to participate in the production of renewable energy transitions such as offshore wind. The next subsection looks at energy transition trends for coastal Heavily Indebted Poor Countries (HIPCs) and raises the question of whether states that have benefited from cumulative historical emissions need to engage in greater financial efforts to support energy access for poor states as a reparation owed to low-income coastal states, particularly the HIPCs, for failure to mitigate emissions in a timely fashion. The final section in this part offers a proposal for pursuing debt relief for clean ocean energy to address the gap in renewable energy infrastructure for many highly indebted states and to pay back some portion of their multi-decade climate debt.

### 3.1. Existing Offshore Wind Sector and Existing Renewable Energy Production for Low and Lower-Middle-Income Coastal Countries

In 2013, the International Renewable Energy Agency calculated offshore wind generation of 7171 megawatts (MW) capacity and 14,535 gigawatt-hours (GWh) of production [17]. By 2022, this number had grown substantially to 54,257 MW of production with approximately 137,614 GWh [17]. As of August 2023, 4C Offshore, a consultancy that collects offshore wind farm statistics, listed 63,081 MW of operational global offshore wind projects across Europe, Asia–Pacific, and the Americas [18]. Wind farms are becoming increasingly large as the cost of electricity associated with offshore wind has decreased. In 2023, the world's largest offshore wind farm was a 1.3 GW wind farm 55 miles (89 km) off the Yorkshire coast in the United Kingdom operated by Danish-based Ørsted with 165 wind turbines that can deliver power to 1.4 million UK homes [19]. In addition to fixed support wind farms, there are also floating wind farms that take advantage of energy production in deep ocean waters. The largest operational floating offshore wind farm in the world is off Norway which at full build-out by the end of 2023 will produce 88 MW of power [20]. The United States is likewise depending on floating wind technology to install off the coast of California as part of the nation's Floating Offshore Wind Shot [21].

The trends in the increasing number of wind farms and ever larger wind farms reflect steady growth in public and private offshore wind investment, albeit not at the ambitious rates suggested by IRENA in 2021 of 382 GW of production by 2030 [1]. Looking towards a global energy transition, the United Nations Conference on Trade and Development in its 2023 World Investment Report observed that renewable power generation is likely to require involvement from international investors because many countries do not have the financial resources for capital-intensive infrastructure [4]. The current investment in the offshore wind sector is not evenly distributed. Much of this investment is in the European region where several coastal states including the United Kingdom, France, and Germany have rapidly expanded their offshore wind capacity. In 2020, the European Union set a target of 60 GW of offshore wind generation by 2030 and 300 GW by 2050 [22]. Achieving Europe's target will mean "multiplying the capacity for offshore renewable energy by nearly 30 times by 2050" and investing "up to EUR 800 billion" [22]. Investments continue in the sector with offshore wind continuing to have the third-largest investment share among all renewables since 2014 [3].

Financial and industry trends point to the rapid expansion of offshore wind expansion, but as noted earlier, the investments are unevenly distributed. For purposes of understanding the difference in access to large-scale ocean renewables by states depending on the income level of a state, it is valuable to compare the top five global offshore wind producers as measured by energy production to all the coastal nations identified by the World Bank as low income (gross national income per capita USD 1135 or less) and lower-middle-income (gross national income per capita USD 1136–4465) (the low-income and lower-middle income coastal states compared in this paper are Algeria, Angola, Bangladesh, Benin*, Cabo Verde, Cameroon*, Comoros*, Cote d'Ivoire*, Democratic Republic of the Congo*, Djibouti, Egypt, Eritrea, The Gambia*, Ghana, Guinea*, Guinea-Bissau*, Guyana*, Haiti*, Honduras*,

India, Iran, Jordan, Kiribati, Lebanon, Liberia, Madagascar, Myanmar, Nicaragua*, Nigeria, Pakitan, the Philippines, Repubic of the Congo*, Samoa, São Tome and Principe*, Senegal*, Sierra Leone*, Solomon Islands, Somalia*, Sri Lanka, Sudan, Syria, Tanzania*, Timor-Leste, Togo*, Tunisia, Ukraine, Vanuatu, Venezuela, Vietnam, Ukraine, and Yemen. Countries with a "*" have qualified for debt relief as Heavily Indebted Poor Countries (HIPCs) and are eligible for the Multilateral Debt Relief Initiative managed by the World Bank, International Monetary Fund, and other institutional creditors). Existing offshore wind production and capacity data used in Table 1 and then for comparing renewable energy production with low-income and lower-middle income states were compiled from the International Renewable Energy Agency's 2023 report on global renewable energy.

**Table 1.** Offshore Wind Production and Total Renewable Energy in Top Offshore Wind-Producing States (2023).

| Country | Offshore Wind Energy Production 2021 gWh [17] | Offshore Wind Capacity 2022 MW [17] | Offshore Wind Farms Operational as of August 2023 [18] | Planned Offshore Wind Projects as of August 2023 [18] | Total Renewable Energy 2021 Production (gWh) [17] | Population (Millions) [23] |
|---------|------|------|------|------|------|------|
| China | 52,711 | 30,460 | 136 | 304 | 2,405,538 | 1,425,887,337 |
| United Kingdom | 35,510 | 13,928 | 43 | 182 | 122,178 | 67,508,936 |
| Germany | 24,375 | 8129 | 29 | 163 | 230,800 | 83,369,843 |
| Netherlands | 7952 | 2571 | 10 | 121 | 40,471 | 17,618,299 |
| Denmark | 7593 | 2306 | 15 | 107 | 26,096 | 5,882,261 |

In contrast to the 5 major offshore wind production states, as of September 2023, only Vietnam among the 53 low-income or lower-middle-income coastal states produces offshore wind. Vietnam had 577 2021 gWh of offshore wind energy production in 2021 [17], 99 MW of offshore wind capacity in 2022 [17], and plans for 114 additional wind farms in addition to the existing 28 farms in 2023 [18]. India has 40 planned offshore projects [18]. No other low-income or lower-middle-income coastal state has public plans for offshore wind or other marine renewable technologies for domestic energy production.

Offshore wind energy is not equally distributed across coastlines but is instead, except for Vietnam, exclusively in countries that have middle to high incomes. Notably, most of the low-income or lower-middle-income coastal states have limited access to renewable energy. While the HIPCs that have a coastline have a total of 369,800,628 people, they presently only have access to 54,976 gWh of renewable energy production [17,23]. From the global total of 7,857,803 GWh of renewable energy in 2022 [17], the most indebted states have access to 0.69% of the available renewable energy even though these states are home to 4.6% of the global population [23].

This trend of differential energy access to both marine and non-marine renewable energy becomes even more apparent when you examine all of the low-income and lower-middle-income coastal states with a total population of 3,306,884,371 residents having access to only 736,208 GWh of renewable energy production [17,23]. When looking at the almost 7.9 million GWh of renewable energy production, this means that 41% of the global population only has access to 9.4% of global renewable energy [17,23]. This has real implications for sustainability as many of these countries attempt to advance basic development goals, which will include building new energy infrastructure. Almost all of these countries are not part of the "offshore wind" revolution or other renewable energy infrastructure efforts because of a lack of financial and technical capacity. In the next section, this paper examines the lack of per capita energy within HIPCs and what the current trends in energy adoption mean for low and lower-middle-income states to achieve a transition to low-carbon energy resources.

Even though there is sizable interest from the private sector, particularly in recent years, in offshore wind development, private investors are unlikely to seek renewable energy investments in many of the HIPCs not just because of general financial risks but also because of a lack of essential infrastructure to support a renewable energy transition including transmission lines [4]. At present, the majority of loans to the least-developed countries take a long time to financially close, are highly leveraged, and have higher interest premiums [4]. Private investors look for government equity in a project before investing in renewable energy infrastructure within many of these countries. Without private investment opportunities, states must rely heavily on multilateral development banks. As discussed below, many of these countries will continue to face challenges in making clean energy transitions and addressing energy access gaps due to ongoing and often unsustainable sovereign debts.

### 3.2. Energy Transitions for Heavily Indebted Poor Coastal Countries

There is a substantial need to accelerate energy transitions in low and lower-middle-income states by investing in renewable energy. To meet the global goals of achieving a renewable energy transition by 2030, renewable energy capacity has to increase at least three times by 2030. In the Middle East and Africa, installed renewable energy capacity needs to increase 10 times to meet growing needs [4].

Many, albeit not all, of the coastal countries identified by the multilateral financial institutions as low-income and lower-middle-income nations have extremely limited access to energy from any source which has limited economic growth potential. Table 2 lists the per capita energy use, per capita emissions for the Heavily Indebted Poor Coastal Countries, carbon intensity of electricity, and the renewable energy share of electricity production from 2013 and 2021.

**Table 2.** Low Per Capita Energy Use by Heavily Indebted Poor Coastal Countries with Carbon Intensity.

| Coastal State | Per Capita Energy Use kWh (2021) [24] | Per Capita Electricity Use kWh (2021) [25] | Per Capita $CO_2$ Equivalent Emissions (Metric Tons) [26] | Carbon Intensity of Electricity (2021) $gCO_2e$ [27] | Renewable Energy Share of Electricity Production (2013)/(2021) (%GWh) [17] |
|---|---|---|---|---|---|
| Benin | 2485 | 18 | 0.6 | 667 | 0/0.2 |
| Cameroon | 1594 | 296 | 0.4 | 278 | 80.9/79.3 |
| Comoros | 1634 | 170 | 0.4 | 714 | 8.2/0 |
| Cote d' Ivoire | 2371 | 400 | 0.4 | 411 | 21.3/18.7 |
| Democratic Republic of the Congo | 411 | 115 | No information | 25 | 99.9/99 |
| The Gambia | 931 | 114 | 0.2 | 700 | 2.1/1.6 |
| Guinea | 1282 | 205 | 0.3 | 209 | 74.1/88.1 |
| Guinea-Bissau | 677 | 39 | 0.2 | 750 | 0/7.1 |
| Guyana | 13,690 | 1529 | 1.92 | 642 | 8.4/7.1 |
| Haiti | 1031 | 86 | 0.3 | 606 | 13.5/18.2 |
| Honduras | 5087 | 1165 | 0.9 | 375 | 41.9/62.7 |
| Liberia | 1065 | 177 | 0.2 | 304 | 21.3/57.4 |
| Madagascar | 508 | 72 | 0.1 | 483 | 55.2/42.8 |

**Table 2.** *Cont.*

| Coastal State | Per Capita Energy Use kWh (2021) [24] | Per Capita Electricity Use kWh (2021) [25] | Per Capita CO$_2$ Equivalent Emissions (Metric Tons) [26] | Carbon Intensity of Electricity (2021) gCO$_2$e [27] | Renewable Energy Share of Electricity Production (2013)/(2021) (%GWh) [17] |
|---|---|---|---|---|---|
| Mauritania | 3989 | 407 | 0.9 | 527 | 4.8/19.4 |
| Mozambique | 2241 | 620 | 0.2 | 127 | 99.8/94.5 |
| Nicaragua | 4265 | 676 | 0.7 | 354 | 52.4/69.3 |
| Republic of Congo | 2348 | 689 | 1.3 | 396 | 54.2/40.7 |
| São Tomé and Príncipe | 3310 | 448 | 0.6 | 600 | 91.7/6.1 |
| Senegal | 2505 | 333 | 0.6 | 523 | 1.9/12.5 |
| Sierra Leone | 493 | 25 | 0.1 | 48 | 62/75.3 |
| Tanzania | 907 | 129 | 0.2 | 517 | 39.4/42.8 |
| Togo | 1116 | 73 | 0.3 | 460 | 23.8/20.7 |

The sum per capita energy use of all of these 22 nations taken together is 53,940 kWh. A single user in the Netherlands, one of the top five offshore wind producers uses close to the same amount (56,001 kWh) [24]. Other global users are far more profligate in net energy use: United States (78,754 kWh), Canada (102,160 kWh), Singapore (147,085 kWh), and Australia (63,459 kWh) [24]. While there are some outstanding examples of states that produce more than half of their electricity from renewable sources (see, e.g., Cameroon, Democratic Republic of Congo, Guinea, Honduras, Liberia Mozambique, Nicaragua, and Sierra Leone), many HIPCs do not have a large share of renewable energy available for electricity production.

For some states without a substantial level of renewables, the energy available is relatively carbon intensive, as indicated by reading the data in the fourth column illustrating relatively high carbon intensity in conjunction with the first and second columns demonstrating both low energy usage and low electricity usage per capita. For example, in Togo, relatively little energy is used per capita but the carbon intensity for the electricity component of that energy is 460 gCO$_2$e, suggesting reliance on fossil fuel to deliver what minimal electricity is available per capita. North America and Europe, where there is far more access to renewables, had an average carbon intensity for electricity of 339 gCO$_2$e and 278 gCO$_2$e, respectively, in 2021 [27]. In contrast, African states across the continent had an average carbon intensity of 488 gCO$_2$e [27].

The last column in Table 2 indicates the trajectory for the uptake of renewables into electricity production. While most countries have made some progress over the course of a decade, there is a subset of countries that appear to have made little or no progress on developing new renewable sources of energy to tap into, at least, for electricity production. Notable countries where there has been a sizable decline in renewable energy usage of 10% or more are Madagascar, the Republic of Congo, and São Tomé and Principle over the past decade. What all these data taken together suggest is that HIPC coastal states have relatively limited access to energy given their population sizes and many of these states have not made a transition to cleaner energy either in terms of scaling up available clean energy or developing new clean energy sources. The need for more energy availability for most of the HIPCs is imperative to support development objectives, but many of these states appear to have challenges in achieving more energy access for their populations even through conventional sources. Pursuing additional energy access for populations that may not have existing energy access by developing more conventional sources will only exacerbate existing climate warming trends.

Given the financial debt cycle experiences of many of the HIPCs and the challenge of tapping into the renewable energy market, the next section examines the international responsibility of sovereign debt holders whose economies have benefited from unregulated emissions but have not achieved climate mitigation objectives under the United Nations Framework Convention on Climate Change. This section provides data on the percentage shares of carbon emissions for top quota holders of the International Monetary Funds, suggesting that these states have some international responsibility for failing to systematically mitigate emissions under the UNFCCC, and proposes that part of this "climate debt" responsibility can be discharged through sovereign debt relief in exchange for renewable energy investments including offshore wind.

### 3.3. International Sovereign Debt Holders Paying and International Responsibility for Failure to Mitigate

It is widely understood that the globe is 1.1 degrees warmer than historic baselines due to anthropogenic greenhouse gas emissions; we are not on track to reduce our warming by even 2 degrees Celsius and we have a global carbon limit beyond which additional carbon emissions are expected to trigger certain impacts [28]. Given the protection of national interest to develop, it has been a subject of contestation who should cut emissions and by how much. Even as warming continues with emerging impacts, there have been only limited discussions about climate accountability where states responsible for the largest historical proportion of global emissions accept some legal responsibility for having benefited from high emissions but not systematically mitigating recent emissions.

Philosopher Henry Shue argued in the context of hearings by the UN Framework Convention on Climate Change's Subsidiary Body for Scientific and Technological Advice on historical responsibility that states that have benefited from a carbon-intensive development process have obligations to communities who "must be able to emit carbon" in order to achieve development needs. From the perspective of fairness and equity, he argued that states should think about carbon budgets in terms of hypothetical permits for different types of activities. If these permits were broadly distributed in a system where we have a carbon cap limit on new emissions, individuals who have basic development needs have priority in the release of the remaining global emissions that are still available in the global carbon budget [29]. Yet, politically powerful states continue to downplay their national economic enrichment that has been achieved due to the ability over multiple decades to have freely released greenhouse emissions. At present, these states, given the energy needs described above, have done relatively little to assist other states at the frontline of climate impacts which include HIPCs to develop sustainable energy systems.

Article 4(2)(a) of the United Nations Framework Convention on Climate Change obligates countries listed in Annex I, which includes most so-called developed states, to "adopt national policies and take corresponding measures on the mitigation of climate change, by limiting its anthropogenic emissions of greenhouse gases and protecting and enhancing its greenhouse gas sinks and reservoirs" [30]. While all of these Annex I states have adopted national policies under the Paris Agreement, these states have not necessarily limited anthropogenic emissions of greenhouse gas emissions. In the United States, for example, it was not until 2019 that the U.S. emitted less carbon dioxide per million metric tons than it had in 1990. Much better progress in the U.S. has been made on reducing methane and nitrous oxide emissions, but no progress has been made on hydrofluorocarbons, perfluorocarbons, sulfur hexafluoride, or nitrogen trifluoride [31]. States such as the United States are internationally legally responsible for breaching UNFCCC obligations to limit anthropogenic emissions of greenhouse gases and take measures on the mitigation of climate change. Despite adequate knowledge of the harms of climate change to protect the interests of other states, the United States has been negligent in its responsibility to curb emissions in spite of the UNFCCC going into effect in 1994 and the U.S. being a member since 1992. This omission to take action comes with consequences including reparation for injuries through restitution, compensation, and/or satisfaction [32]. Where it may not be

possible to provide restitution for harm, it may still be possible to provide restitution in the form of compensation. One possibility that will be discussed in Section 3.4 is for this compensation for decades of inaction to come in the form of debt relief.

Existing efforts for mitigation based on concepts of historical responsibility have been ineffective. The Kyoto Protocol failed to tackle actual carbon consumption but focused only on carbon production as wealthier countries imported carbon in the form of carbon-intensive goods from non-participants in the Kyoto Protocol regime [33]. The current nod to historical responsibility is the UN Framework Convention on Climate Change's Loss and Damage Fund designed to assist states with adaptation through a variety of proposed financial mechanisms to be funded by pledges from countries and philanthropies including social protection funds, catastrophe risk insurance, and catastrophe bonds [34]. Other proposals for future funding include international taxes, particularly on the fossil fuel industry, and debt for loss and damage swaps [34]. The idea of a variation on the "debt swap" proposed at the 27th UNFCCC Conference of Parties meeting that might address the inequities associated with climate debt will be explored in Section 3.5 of this paper.

Due to ongoing impoverishment triggered in part by large public debt loads, many states, including the HIPCs, are unable to make energy transitions including participating in ongoing global efforts to scale up offshore wind which holds the promise of helping certain coastal HIPCs with sufficient wind resources to become energy independent. It is important to remember that most of the countries with the largest shares of global cumulative carbon emissions are also some of the largest creditors in international financial institutions as illustrated in Table 3 below.

**Table 3.** Top 10 quota holders in the International Monetary Fund, Cumulative $CO_2$ Contributions, and Percentage Global Share in Cumulative Emissions.

| | Quota in the IMF (Percent of Total) [35] | Cumulative $CO_2$ Measured from 1750 (Billion Tons of $CO_2$) [36] | Percentage Share of Global Cumulative $CO_2$ Emissions as of 2021 [37] |
|---|---|---|---|
| United States | 17.43 | 421.91 | 24.29% |
| Japan | 6.47 | 66.71 | 3.84% |
| China | 6.40 | 226.92 | 14.36% |
| Germany | 5.59 | 93.29 | 5.37% |
| France | 4.23 | 39.11 | 2.25% |
| United Kingdom | 4.23 | 78.51 | 4.52% |
| India | 2.75 | 57.11 | 3.29% |
| Russia | 2.71 | 117.55 | 6.77% |
| Brazil | 2.32 | 16.67 | 0.96% |
| Canada | 2.31 | 34.12 | 1.96% |

The top ten International Monetary Fund creditors holding quota are responsible for 67.6% of the global cumulative carbon emissions with most of the cumulative contributions from the United States, China, and Russia.

The lower-middle-income countries and the low-income countries, including both coastal and non-coastal states, together account for 10.5% of the cumulative emissions but are home to more than half of the current global population with 3.4 billion individuals in lower-middle-income states and 718.26 million in low-income states [37,38]. While regions such as Africa and the Middle East need to increase their renewable energy capacity by tenfold to meet growing local energy needs, at present, Europe only needs to double its capacity. Both states will need about the same amount of annual investment to achieve these different installation capacities (Africa and Middle East USD 170 billion, Europe USD 180 billion). Europe, however, had potential access to international investment in

2022 of around USD 248 billion while Africa only had potential access to USD 45 billion of investment [4].

A new approach is needed to square the ongoing state responsibility for unmitigated emissions and the need to accelerate energy transitions for the poorest nations that have not achieved domestic energy equity. The legacy of multilateral lending continues to keep some states in a cycle of impoverishment and prevents already disadvantaged states from gaining access to larger-scale low-carbon technologies. A total of 107 out of 147 developing countries have no strategy specifying sources of finance to assist with the energy transition [4]. Many of these countries remain indebted to public debtors which slows the amount of domestic public funding available for renewable energy infrastructure.

*3.4. Need for Debt Relief for Clean Energy Transition*

In the 1970s and 1980s, numerous low-income countries borrowed from governments or export credit agencies who would accept risks of non-repayment. At that time, export credit guarantees from lending countries were considered to be complementary to official development aid as a means of stimulating economies [39]. Much multilateral diplomatic attention in the last several decades has gone to addressing the debt crisis of low-income countries, many of these countries emerging from recent decolonization. Early attempts to relieve debt involved rescheduling repayments and refinancing loans. Rescheduling of debt service payments, however, did not eliminate the debt but increased scheduled debt service because of additional interest and principal payments associated with rescheduling [39]. Creditor agencies have been reluctant to reduce national debts although rescheduling agreements have forgiven some debt and rescheduled other debt at low interest rates. The International Monetary Fund and the World Bank initiated the Heavily Indebted Poor Country Initiative in 1996 to try and address unmanageable debt burdens for 39 developing countries that are only eligible for highly concessional aid and who have demonstrated some intention to reform policies including developing a Poverty Reduction Strategy. The initiative relieved many of the qualifying states of billions of dollars of debt [40]. Table 4 illustrates that sizable debt levels continue for many energy-poor states. Most of the HIPCs carry high levels of external public debt constituting for many of these countries around one-fifth of their GDP. Other countries in the world including China and the United States also carry high levels of debt but this debt is not in the form of external bilateral or multilateral public debt.

**Table 4.** List of Heavily Indebted Poor Coastal Countries with External Debt, Percentage of Debt from Multilateral/Bilateral Institutions, Percentage of Public Debt as a share of GDP, Per Capita external public debt, and availability of offshore oil and gas resources.

| State | Amount of External Debt (Billions) [41] | Percentage of External Debt as Multilateral and Bilateral Financial Institution Debt [41] | Percentage of External Public Debt as a Share of GDP [41] | External Public Debt per Capita [41] | Availability of Offshore Oil or Gas [42–44] |
|---|---|---|---|---|---|
| Benin | 6 | 57 | 32.9 | 467 | Yes |
| Cameroon | 12 | 83.5 | 27.4 | 456 | Yes |
| Comoros | <1 | 100 | 21.2 | 296 | No |
| Cote d' Ivoire | 23 | 43.5 | 32.3 | 838 | Yes |
| Democratic Republic of the Congo | 7 | 70.9 | 11.5 | 70 | Yes |

**Table 4.** *Cont.*

| State | Amount of External Debt (Billions) [41] | Percentage of External Debt as Multilateral and Bilateral Financial Institution Debt [41] | Percentage of External Public Debt as a Share of GDP [41] | External Public Debt per Capita [41] | Availability of Offshore Oil or Gas [42–44] |
|---|---|---|---|---|---|
| The Gambia | 1 | 90 | 39.7 | 324 | Yes |
| Guinea | 4 | 92.1 | 22.2 | 251 | No |
| Guinea-Bissau | 1 | 66.1 | 55.1 | 512 | Yes |
| Guyana | 1 | 97.8 | 17.5 | 2000 | Yes |
| Haiti | 2 | 98.1 | 10 | 176 | No |
| Honduras | 9 | 72.5 | 30.5 | 853 | No |
| Liberia | 1 | 100 | 28.9 | 196 | No |
| Madagascar | 4 | 96.9 | 25.5 | 132 | Yes |
| Mauritania | 4 | 100 | 40.8 | 951 | Yes |
| Mozambique | 11 | 88 | 67 | 330 | |
| Nicaragua | 6 | 99.6 | 45.8 | 980 | No |
| Republic of Congo | 6 | 62.2 | 51.9 | 1000 | Yes |
| São Tomé and Príncipe | <1 | 95.8 | 44.1 | 1000 | No |
| Senegal | 14 | 66 | 52.2 | 839 | Yes |
| Sierra Leone | 1 | 87.2 | 31.6 | 161 | No |
| Tanzania | 19 | 81.8 | 27 | 317 | Yes |
| Togo | 2 | 83.8 | 20.9 | 208 | Yes |

In June 2023, the World Bank agreed to suspend debt payments on new loans with Climate Resilient Debt Clauses in the case of extreme weather events, including events exacerbated by climate change [45]. These new loan clauses are, however, not intended to operate as debt forgiveness but instead to provide flexibility in debt repayment. These clauses may leave countries when repayments resume in potentially as vulnerable a financial position as before the disaster event triggering the clause. To pay off debts, some states may seek to pursue conventional offshore energy development. Table 4 also illustrates that most of the HIPCs have some access to offshore oil and gas reserves. Not all of these reserves are currently in production.

From a perspective of deep rather than shallow sustainability, there is a question of whether it might be possible to develop a different approach to dealing with existing public debt loads and energy transitions so that HIPCs can take a different development pathway from developing conventional energy supplies. Sustainable Development Goal funding has unfortunately only made a minimal difference for many of the HIPCs in increasing renewable energy consumption. In Table 5, the capacity of existing country-wide renewable energy for HIPCs is compared in 2016 when the Sustainable Development Goals were adopted with the most recent data from 2022. The proportion of renewable energy in final energy consumption between 2000 and 2020 is also compared to observe general trends in the uptake of renewable energy.

**Table 5.** Changes in HIPC Installed Renewable Capacity and Renewable Energy Share over time.

| State | Installed Renewable Energy Capacity (Watts per Capita) 2016 [17] | Installed Renewable Energy Capacity (Watts per Capita) 2022 [17] | Percentage Renewable Energy Share in Total Final Energy Consumption (2000) [46] | Percentage Renewable Energy Share in Total Final Energy Consumption (2020) [46] |
|---|---|---|---|---|
| Benin | 0.3 | 2 | 70.29 | 46.2 |
| Cameroon | 31 | 30 | 84.59 | 78.94 |
| Comoros | 1.8 | 2 | 69.87 | 48.29 |
| Cote d'Ivoire | 23 | 43.5 | 63.72 | 63.34 |
| Democratic Republic of the Congo | 32 | 30 | 97.94 | 96.16 |
| The Gambia | 1.6 | 1 | 62.86 | 49.74 |
| Guinea | 33 | 62 | 85.52 | 65.77 |
| Guinea-Bissau | .2 | 1 | 91.24 | 87.22 |
| Guyana | 60 | 67 | 30.56 | 12.04 |
| Haiti | 5.4 | 7 | 80.56 | 76.31 |
| Honduras | 158 | 191 | 55.24 | 50.09 |
| Liberia | 5.7 | 18 | 91.34 | 92.96 |
| Madagascar | 7 | 7 | 82.17 | 84.75 |
| Mauritania | 17 | 26 | 44.41 | 23.78 |
| Mozambique | 80 | 72 | 93.64 | 80.91 |
| Nicaragua | 105 | 111 | 58.42 | 52.13 |
| Republic of Congo | 43 | 40 | 64.86 | 71.88 |
| São Tomé and Príncipe | 12 | 12 | 54.73 | 41.61 |
| Senegal | 4.5 | 26 | 47.52 | 38.64 |
| Sierra Leone | 12 | 13 | 93.32 | 75.07 |
| Tanzania | 13 | 11 | 93.73 | 83.95 |
| Togo | 9.2 | 15 | 77.11 | 76.62 |

Given an increase in Sustainable Development Goal Funding, policymakers would expect the trend to be a sizable increase in installed renewable energy capacity but the trends for the HIPCs either show little growth in capacity or even a small decrease in renewable capacity due to potentially non-operational energy infrastructure. This is in contrast to renewable energy capacity numbers for European and North American states that demonstrate a positive absolute change from 2016 to 2020 [17]. This inertia has implications for development, climate mitigation, and climate adaptation.

*3.5. Proposal for a Debt Relief Program in Exchange for Investments in Renewable Ocean Energy*

Current proposals to address the impact on states that are likely to experience climate impacts include a proposal to remove sovereign debt as a form of "loss and damage" payments when a climate-related disaster triggers specific loss and damage [34]. This approach is "too little too late" and ignores the structural fragility of already impoverished countries. While an influx of post-disaster funds can provide immediate and needed relief, this funding will be unlikely to address the larger infrastructure needs that will assist states in making transformative changes to a low-carbon economy.

Instead of debt relief for climate harms, debt relief exchanges can instead be used strategically and proactively to assist publicly indebted coastal states in making long-term investments in low-carbon energy futures. A debt relief program will avoid the problem of waiting on donor countries to exercise political will to provide sufficient funding to multilateral mechanisms. In the recent past, waiting for donor countries to step forward to help other states achieve low-carbon futures through financing has not been an effective strategy for those states who wanted to make a low-carbon transition but could not afford the economic opportunity costs of foregoing carbon-intensive resource development. A good example of this is the Yasuni-ITT project where the Ecuadorian government committed in 2007 to not drill in the Ishpingo-Tambococha-Tiputini oilfield in Yasuni National Park in exchange for international donations that would offset the expected revenue from the oil extraction activities. The donations never materialized and the Ecuadorian government proceeded in 2016 with authorizing drilling in about 0.01% of the park with a daily production of 55,000 barrels a day [47]. Seven years later, in a 2023 referendum, Ecuadorian citizens voted by a majority to ban oil drilling in the region [47]; Ecuador's oil industry had, however, already contributed an additional annual carbon load of about 8,553,957,500 $kgCO_2$ (8 million metric tons of carbon) that would have been avoided if international donors had given some assistance to Ecuador to cover opportunity costs of not drilling.

Debt swaps, particularly debt reduction for conservation work projects, have been more successful than Ecuador's efforts to solicit international donations. Chamon, Klok, Thakoor, and Zettelmeyer identified over 140 debt swaps including tripartite debt swaps involving an NGO or a new lender and bilateral debt swaps [12]. Most of these programs were relatively small in value with a total value in 2017 of USD 2.6 billion. In many cases, these programs simply replaced old debt with new debt [12].

### 3.5.1. Structure for Debt Relief for Energy Program

A debt relief program would be structured with a debtor state through either a bilateral or multilateral agreement. Figure 1 illustrates a basic design for a "debt relief for clean ocean energy program". The creditor country or multilateral bank could address its "climate debt" arising from state responsibility through the elimination of debt service payments. These payments would serve as reparation for the omission of adopting effective emission reduction policies and continuing to benefit from high-emission economies. Assuming that a particular state has a reasonable potential for ocean energy production, some pre-negotiated portion of the debt service payments would be invested in siting, building, and managing ocean clean energy infrastructure projects. Where a state has renewable energy development capacity but not expertise, the investment will probably result in joint projects between states, companies, and other private stakeholders (universities) with existing ocean renewable energy development skills. To protect clean energy investment funds from potential theft, the parties could agree that monies previously used for debt service payments would be transferred to an independent escrow account to be released for specific types of mutually agreed upon climate projects. If the funds are not used within a negotiated time period for a state to find a joint partner or undertake the project on its own, the funds would revert to the creditor.

This proposed ocean-focused energy infrastructure program would work well in parallel with recent proposals to forgive debt in exchange for countries investing in adaptation efforts so that climate-vulnerable states can make critical risk reduction investments now [48], to leave undeveloped and unassigned gas reserves in the ground for 10 years [49], and to reduce debt service payments for countries in the Amazon that effectively reduce national deforestation rates [50]. The proposal here for waiving debt payments would ensure a similar measurable outcome but would offer a different approach by triggering the construction and operation of a single infrastructure project. It would offer states an opportunity to reallocate money that might otherwise be used for debt payments into discrete renewable projects designed for a reasonable operational timeline for a large in-

frastructure project (e.g., 10–12 years). The amount of debt forgiveness would depend on the costs associated with the project which would include training costs to ensure that the development and operation of the project does not become entirely outsourced from the region.

**Figure 1.** Process for a "Debt Relief for Clean Ocean Energy Program".

### 3.5.2. Justification for Debt Relief for Ocean Clean Energy Program

There is a strong interest in seeking investment in offshore wind among several low and middle-income countries even though it is costly infrastructure. In 2019, the World Bank identified the potential for offshore wind construction in low and middle-income countries by evaluating access to sizable financing (USD 10–50 million in costs), plans to create an electricity grid, early investments (projects will take 5–10 years to bring to operation), and regional cooperation [51]. The 2019 report observed that Brazil, India, Morocco, the Philippines, South Africa, Sri Lanka, Turkey, and Vietnam together have the offshore wind potential to generate 3082 GW of energy [51]. This is nearly equivalent to all of the renewable energy capacity available in 2022 [17]. The only country that has actual operational offshore wind based on early investments and electricity grid planning as of 2023 is Vietnam. Renewable ocean energy for other states such as the HIPCs discussed in the sections above, which often carry unsustainable levels of debt, has not been discussed by international policymakers.

Yet, with sovereign debt relief, states such as the HIPCs may be able to make investments that would otherwise be unattainable. In addition to the low and middle-income countries identified by the World Bank as promising sites for wind development, some coastal states such as Mauritania, based on readings from the Global Wind Atlas (https://globalwindatlas.info/en/area/Mauritania, accessed on 22 August 2023), have potential high offshore wind rates comparable to the coasts of China and Denmark. Some investors have recognized this potential and are seeking to install a "green hydrogen" hub to develop hydrogen-based fuels for export [52]. While this may have an overall positive development on Mauritania's economic development, it does not directly address Mauritania's needs for its own energy transition.

For a country such as Mauritania with USD 4 billion in external debt mostly from multilateral and bilateral aid, Mauritania might work with its creditors to reduce its debts by building an 83.12-MW turbine floating windfarm to generate 1 GW. To construct such a wind farm, assuming development and project management, would be around USD 10.1 million per turbine plus operational costs of UDS 1 million per turbine for 30 years lifespan of a wind farm and decommissioning of around a half million per turbine [53,54]. This would be around USD 2.9 billion over the course of the life of a wind farm. A USD 3 billion debt relief package in exchange for an operational wind farm may help Mauritania make an energy transition. As Table 5 indicates, today Mauritania has a smaller percentage of energy produced from renewables in 2020 than it did in 2000, suggesting that Mauritania may also be relying on additional fossil fuel investments to meet its energy demands. One gigawatt of production from a wind farm would meet all of Mauritania's residential electricity needs. Analogizing from the U.S. Department of Interior statistics that a U.S. household of 4 uses 10,655 kw/H and that 1 GW of wind power could supply at least 225,000 such homes [55], the average 4-person household in Mauritania would use 1628 kw/h so that 1 GW of energy would power 1.4 million households and there would still be surplus energy to bring energy to those without energy resources or to invest for other national priorities. Other ocean-based clean energy strategies such as wave turbines or green hydrogen production from seawater might also benefit from investment funds made available after debt relief.

Debt relief for climate action offers a reasonable mechanism for financing investments that are currently financially unattractive for private investors. With sovereign debt relief, creditor governments are repaying some of their climate debt that has accrued as part of state responsibility for failing to meet UNFCCC mitigation obligations. Debtor nations benefit from the potential for achieving energy independence for states. "Debt Forgiveness for Clean Ocean Energy" deals would contribute to energy sovereignty for coastal states by allowing these states to make capital investments to shift to supply energy into their own national and regional markets rather than into the global markets for oil and gas development. One of the promises of renewable energy from both a sovereignty and sustainability perspective will be the localization of clean energy production for national development objectives.

Five aspects of the "Debt Relief for Clean Ocean Energy" proposal are significant. First, this proposal is a legitimate approach to addressing the climate debt created by certain states that have benefited from contributing the most to the cumulative impacts of greenhouse gas emissions. It offers a just transition to a sustainable pathway for nations and regions that might otherwise not be able to capitalize on marine clean energy resources in any meaningful fashion. Second, this proposal scales up financing and restores political sustainability by eliminating what has become inescapable debt. Third, it opens up the opportunity for states to focus future national energy development in spaces where there are less likely to be conflicts with communities over the protection of land for food production or conflicts over terrestrial habitat protection. Acknowledging that developing and operating ocean renewable energy projects can generate conflicts among ocean space users, future projects need to be carefully designed in partnership with stakeholders to protect existing and potentially competing ocean uses including habitat uses, fishing, and shipping. Fourth, the project approach facilitates states receiving needed technology transfers with the political support of key members of the global financial community. Finally, designating debt relief money for specific ocean energy projects will make it easier to measure progress toward specific goals of enhancing energy access while improving ocean conservation. The goals of a clean energy production infrastructure project in contrast to a flexible climate adaptation program will be less open to interpretation.

Pursuing any debt relief package requires a detailed analysis of numerous factors including how much debt would be removed, how many fiscal resources would actually be allocated to ocean energy investment, how to involve stakeholders, and how a debt relief package might otherwise impact a country's balance of payments [56]. There is no

single model debt relief package, but the general idea is for those creditor states who have been enriched by climate emissions and who have failed to mitigate in keeping with their UNFCCC obligation to accept responsibility and compensate by giving debt relief that will enable otherwise heavily indebted states to invest in clean energy investments.

### 3.5.3. Limitations of Debt Relief for Clean Ocean Energy Program

Critics may argue that the need for the exercise of "generous" political will on the part of creditor nations would make these types of debt relief packages a financial non-starter. While it is true that creditors need to act first to make any debt relief package a reality because the power in the creditor–debtor relationship lies with the creditor, it is worth noting that the monies that would be involved in any debt relief program have already been disbursed from multilateral banks or under bilateral agreements as part of previous loans. Creditors are not being asked for new investments but instead to forego a financial repayment as a signal of recognizing state responsibility for a failure to systematically mitigate emissions. The financial creditor community has already demonstrated some appetite for debt relief through programs such as the 1996 Heavily Indebted Poor Countries Initiative and the 2005 Multilateral Debt Relief Initiative. There is also recent interest expressed by development agencies in creditor nations in deploying some form of "debt swap" for climate financing [16]. These programs, however, have not yet led to comprehensive debt relief, and more efforts are needed. What these programs and agency discussions indicate is some appetite for addressing chronic indebtedness using more innovative tools.

The proposed program "debt relief for clean ocean energy" will not lead to comprehensive debt relief either but would be another step in the right direction. A major challenge for a program will be to ensure that there is sufficient money available from expected debt service payments in order to populate an escrow fund. For some countries already struggling with repayment schedules, it may not be possible to amass sufficient money from redirecting debt payments, and in these cases, it may be better for a country to seek grants. Even for countries where there is debt service money available for energy investment, it will be critical that money in an account is used for renewable energy infrastructure projects and not for other general purposes, which can be tempting given that several heavily indebted states have multiple pressing development needs. Where there is adequate funding available, ensuring that projects are well managed will require adequate domestic technical capacity for financial and project management. For some states, this may prove to be a challenge particularly where there may already be existing governance challenges with combatting financial corruption. One protection that is contemplated in the proposed debt-relief scheme described above is the creation of an escrow account that would limit what disbursements are possible for project expenses.

Some creditor states may prefer to deliver conditional climate grants because they are perceived to be a safer investment in climate projects than debt relief packages [12]. Conditional grants may have fewer transactional costs than a sovereign debt restructuring effort and grants may provide sufficient upfront costs to allow for initial energy investments for a country that is regularly struggling with meeting basic debt service payments. Even with potentially more overall transactional costs, the debt relief proposal, for those countries with sufficient funding freed up from the debt service, has the distinction of giving some agency back to the debtor country over domestic energy planning independent of the preferences of donor countries. A former debtor country that is investing its own money in energy development may generate more national pride and ownership in ongoing energy and sustainability projects.

The problem with relying on conditional climate grants is similar to the overall challenge of a debt relief package. Grants depend on the willpower of creditor nations, which has been weak. If creditors refuse debt relief, many states will not be able to implement Nationally Determined Contributions under the Paris Agreement [12]. If debt relief is not available, indebted states will have to rely on grants which have not been forthcoming. Global climate finance gaps have been a chronic problem for developing countries with

donor countries chronically falling short of meeting the 2009 promise of USD 100 billion annually for mitigation and adaptation for low- and middle-income countries. According to recent reports by investigative journalists, some of this money has apparently been spent in unusual ways including expanding ice cream shops in Asia rather than furthering local basic development [57].

Any proposal such as the "debt relief for ocean energy" proposal in this paper will have financial implications on the economy. A detailed analysis of the financial implications of debt relief proposals on a specific economy is beyond the scope of this paper. It is worth noting, however, that general financial impacts from a policy involving a proposed debt relief package may have political consequences. How a specific government agrees to spend money that is made available under a debt relief package may generate domestic controversy when there are many demands within an already constrained economy. Citizens in a HIPC may have preferences for other priorities beyond large-scale clean energy production. These citizen preferences may be expressed in votes of no-confidence for administrations that negotiate a "debt-relief for clean energy" package when a majority of citizens would prefer to see government funding invested in other development efforts perceived to be more immediate. What this might mean for a debtor country entering into potential debt relief efforts is the need for broad community engagement to ensure that a debt relief effort is domestically viable.

This proposal is not naïve in recognizing that 1 GW wind farms installed in a handful of coastal countries will not offset the pre-existing emissions from more numerous financial creditor states. Much more effort needs to be made by "climate debtor" countries to make amends for continued harm driven by largely unregulated carbon and carbon-equivalent emissions. What is, however, important with the debt relief proposal is that it provides states that have been largely discounted as partners in low-carbon transition the ability to become lead players in national and regional energy transitions. It is both a step for "climate debtor" states to accept responsibility and also for creating conditions within sovereign debtor states for equitable large-scale energy development. If nations genuinely believe in "energy for all" as an equity principle, why should a heavily indebted poor country be the last to have access to large-scale and efficient clean energy?

## 4. Conclusions

As nations with financial capital built in part from financial enrichment generated by decades of carbon emissions pursue large-scale clean energy solutions, many of the poorest states are still not able to participate in the possibilities of producing large-scale and low-carbon energy. Poorer states that have contributed the least to the current climate peril have few options available to them as part of a clean energy transition due to their ongoing indebtedness to international creditors including public creditors. The most indebted coastal states have access to 0.69% of the available renewable energy even though these states represent 4.6% of the global population.

Solutions to global energy transitions need to recognize the inequity of imposing unsustainable financial debts on millions of people whose governments cannot invest in clean energy because they have to prioritize debt service payments. This paper argues that states that have failed to reduce emissions over the course of decades in compliance with UNFCCC obligations to "adopt national policies and take corresponding measures on the mitigation of climate change" have a state responsibility to remedy their impacts on making a systematic energy transition. One remedy includes providing debt relief that will not lead to more debt restructuring but rather investing debt service payments into clean energy investments. This paper has contributed a "debt relief for clean energy" proposal to waive public debt payments owed to multilateral and bilateral financial institutions in exchange for national commitments from low-income states to make strategic long-term clean energy investments including, where appropriate, ocean energy investments. This paper builds on previous calls from researchers to scale up debt-for-climate swaps [11–14]

and extends the concept to specific large-scale infrastructure efforts that will assist with the energy transition.

With so many countries in endless cycles of debt and with the need to accelerate a global energy transformation, achieving sustainability requires immediate investments not only to allow states to rebuild their social capital as debt-free states but also financial capital to fund capital-intensive startup costs associated with clean energy infrastructure. Achieving these outcomes will only be possible for many states through debt relief that provides states with the opportunity to make long-term investments in sustainable energy infrastructure that has been unattainable under current debt loads.

**Funding:** This research received no external funding.

**Institutional Review Board Statement:** Not applicable.

**Informed Consent Statement:** Not applicable.

**Data Availability Statement:** Data used in this report came from the following publicly available datasets: (1) International Renewable Energy Agency, Renewable Energy Statistics 2023, available online at: https://mc-cd8320d4-36a1-40ac-83cc-3389-cdn-endpoint.azureedge.net/-/media/Files/IRENA/Agency/Publication/2023/Jul/IRENA_Renewable_energy_statistics_2023.pdf?rev=7b2f44c294b84cad9a27fc24949d2134 (accessed on 9 October 2023) (providing data on offshore wind energy production and total renewable energy production, total renewable energy production); (2) Global Offshore Wind Farm Database and Intelligence, available online at: https://www.4coffshore.com/windfarms/#db (accessed on 30 September 2023) (listing numbers of operational and planned offshore wind projects); (3) Our World in Data, Energy Use Per Person, available online at: https://ourworldindata.org/grapher/per-capita-energy-use (accessed on 30 September 2023) which uses data from https://www.eia.gov/opendata/bulkfiles.php (accessed on 30 September 2023); https://www.energyinst.org/statistical-review (accessed on 30 September 2023); https://www.gapminder.org/data/documentation/gd003/; https://population.un.org/wpp/Download/Standard/Population/ (accessed on 30 August 2023); https://dataportaal.pbl.nl/downloads/HYDE/ (accessed on 30 September 2023); https://github.com/open-numbers/ddf{-}{-}gapminder{-}{-}systema_globalis (accessed on 30 September 2023); (4) Our World in Data, Per capita electricity generation, 2022, available online at: https://ourworldindata.org/grapher/per-capita-electricity-generation (accessed on 9 October 2023) which uses data from https://ember-climate.org/data-catalogue/yearly-electricity-data (accessed on 30 September 2023); https://ember-climate.org/insights/research/european-electricity-review-2022/ (accessed on 30 September 2023); https://www.energyinst.org/statistical-review/ (accessed on 30 August 2023); https://www.gapminder.org/data/documentation/gd003 (accessed on 30 September 2023); https://population.un.org/wpp/Download/Standard/Population (accessed on 30 September 2023); https://dataportaal.pbl.nl/downloads/HYDE/ (accessed on 30 September 2023); https://github.com/open-numbers/ddf{-}{-}gapminder{-}{-}systema_globalis (accessed on 30 September 2023); (5) World Bank, $CO_2$ Emissions (metric tons per Capita), available online at: https://data.worldbank.org/indicator/EN.ATM.CO2E.PC (providing data on carbon emissions); (6) Our World in Data, Carbon Intensity of electricity per kWh, available online at https://ourworldindata.org/grapher/carbon-intensity-electricity (accessed on 30 September 2023), which uses data from https://ember-climate.org/data-catalogue/yearly-electricity-data/ (accessed on 30 September 2023); https://ember-climate.org/insights/research/european-electricity-review-2022/ (accessed on 30 September 2023); https://www.energyinst.org/statistical-review/ (accessed on 30 September 2023); (7) IMF Members' Quotas and Voting Power, and IMF Board of Governors (21 August 2023), available online at: https://www.imf.org/en/About/executive-board/members-quotas#G (Providing data on country quotas) (accessed on 30 September 2023); (8) Our World in Data, Cumulative $CO_2$ emissions available online at: https://ourworldindata.org/grapher/cumulative-co-emissions, which uses data from www.globalcarbonbudget.org (accessed 30 September 2023); (9) Our World in Data, Cumulative Share of $CO_2$ emissions by world region available online at: https://ourworldindata.org/grapher/cumulative-co2-emissions-region, which uses data from which uses data from https://globalcarbonbudget.org (accessed 30 September 2023); (10) UN Conference on Trade and Development, World of Debt Dashboard, available online at: https://unctad.org/publication/world-of-debt/dashboard# (accessed on 30 August 2023) (providing data on amount of external debt, percentage of debt as multilateral and bilateral financial institution debt, percentage of external public debt as a share of GDP, and external public debt

per capita); and (11) Our World in Data, Sustainable Development Goal Indicator 7.2.1, available online at: https://ourworldindata.org/sdgs/affordable-clean-energy (accessed 30 September 2023) which uses data from https://unstats.un.org/sdgs/metadata/files/Metadata-07-02-01.pdf (accessed 30 September 2023).

**Conflicts of Interest:** The author declares no conflict of interest.

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
