# Peer review of "Multilateral Debt Relief for Clean Ocean Energy"

_sustainability, doi:10.3390/su152014702_

Round 1

Reviewer 1 Report

The main purpose of this paper is to highlight the concept of “climate debt” and argue that it is incumbent on international development banks to invest in debt forgiveness-for-clean energy swaps to allow for both equitable development and future energy sustainability. The concepts involved in the topic are cutting-edge and of writing value. The paper collected a number of publicly available datasets and then further analyzed it. The paper summarized the issues existing in current situation of policy and practice, and finally contributed a proposal. The writing style of the article is skilled, which reflects the professional degree of the author in terms of references and so on. In general, the conclusion could justify itself.

This note is timely and well researched. I certainly read with great interests. However, I still have some suggestions to improve the overall quality of this paper.

1. In the process of data analysis, it is necessary to describe the reasons for the selection of data, as well as the sources of these data. For example, why choose the data for Offshore Wind Energy Production and Capacity 2021 and not for other years?

2. In terms of writing illustrations, under heading 3.1, offshore wind revolution and renewable energy access are juxtaposed. Why does this article only show the relevant data of offshore wind revolution in tables while the date of renewable energy access is not on displayed?

3. In terms of writing length, the third part of this article uses a detailed table to list the data. However, there is a mechanical duplication between the text content and the table content. It is recommended to simplify and highly summarize the content of the text which is mechanically repeated with the tabular information. The author could consider behind the surface of the data to carry out in-depth theoretical analysis of the reasons for the data.

4. In terms of academic norms, some parts of the article contain typographical errors, such as lines 78 and 79 under heading 3.1. It is suggested to check the full text and correst the relevant mistakes.

Author Response

I am very grateful to the reviewer for their constructive comments on this paper.

  1. On the comment  of how data was selected. I have made significant revisions that are reflected the methodology paragraph. I am using the most recent data available from International Renewable Energy Agency in its 2023 report on global renewable energy- this report uses 2021 data for Production and 2022 data for capacity. I have looked elsewhere for public datasets that are more up to date but have not been able to locate any more recent data.
  2. I have changed the heading to “existing offshore wind sector” because I think that it better describes what I am trying to illustrate in this section. I am hoping to make it very clear that many high population states do not have much access to renewable energy and no access to offshore wind infrastructure in spite of having coastal proximity.
  1. Thank you for the comments on the multiple tables that form the basis for some of the observations in the text.  I know that there are many tables in the article but I thought it was important to see if the data could speak for itself in so far as illustrating the gaps between the “haves” and “have nots” of renewable energy—most of the "have nots" are also States carrying unsustainable levels of public debt.  What I have tried to do with all the tables is to give readers the chance to see some of the background trends that are the reason for the proposal being made in this paper. The trends that I am trying to highlight are lack of renewable energy among large portion of the global population (Table 1), the low per capita energy use by the poorest countries with a reasonably high level of carbon intensity (Table 2), the heavy financial debt loads of many of the countries that might wish to invest in lower carbon energy (Table 4),  and the apparent slowdown in uptake of renewable energy for many of the poorest countries(Table 5).  While in some respects, these trends are not surprising- they provide more concrete data-driven evidence of why a proposal to accelerate clean energy investment is needed particularly for the heavily indebted states who may be understandably tempted to  invest in known conventional energy sources rather than emerging renewables .
  1. Thank you for noting typographical errors.  I have gone through the manuscript and made editorial corrections

Reviewer 2 Report

The author discusses the issue of differential access to renewable energy resources, particularly in Heavily Indebted Poor Countries (HIPC) with coastlines. It highlights the disparity in renewable energy access, pointing out that these countries, which represent 4.6% of the global population, only have access to 0.69% of the available renewable energy. The article also discusses the challenges faced by low and lower-middle-income coastal states in transitioning to low-carbon energy resources and the concept of climate debt.

The article presents several important findings and raises significant concerns regarding energy access and sustainability in HIPC coastal countries. It emphasizes the need for a more equitable distribution of renewable energy resources and highlights the challenges faced by these countries in advancing their development goals due to limited energy access. The data presented in the article, including per capita energy use, emissions, and renewable energy capacity, help illustrate the disparities in energy availability.

The article suggests a novel approach to addressing climate debt by proposing "Debt Forgiveness for Clean Ocean Energy" swaps. This proposal entails forgiving a portion of a country's external debt in exchange for the country's commitment to investing in clean ocean energy projects, such as offshore wind farms or wave turbines. The aim is to help these countries transition to low-carbon energy sources and reduce their carbon emissions.

The article's strengths include its in-depth analysis of energy access disparities, its proposal for addressing climate debt, and its use of data to support its arguments. However, there are several areas where the article could be improved:

Contextualization: While the article provides substantial data, it could benefit from more contextualization and explanation of key concepts. This would help readers, especially those unfamiliar with climate debt and energy access issues, better understand the significance of the findings.

Clarity and Structure: The article's structure could be enhanced for better readability. It should have clear section headings and subheadings to guide readers through the complex issues discussed.

Policy Implications: The article touches on potential policy implications but could delve deeper into the practicalities and challenges of implementing debt forgiveness for clean energy projects. It should also discuss potential objections or limitations to the proposed approach.

Case Studies: Including specific case studies or examples of countries that have successfully implemented similar debt-for-energy swaps would provide valuable insights and real-world applications of the proposed concept.

Conclusion: The article lacks a clear concluding section summarizing key points and reinforcing the significance of the proposed approach.

The article addresses an important issue related to energy access disparities and climate debt. While it presents a promising proposal for debt forgiveness in exchange for clean energy investments, it would benefit from improved clarity, context, and policy analysis to make its argument more compelling to a broader audience.

Author Response

Thank you for your very helpful and constructive comments. They have helped me to edit this piece and hopefully provide more clarity on my major points.

Contextualization-

This was a very helpful comment and I have added some additional discussion on international State responsibility as a legal rule to help explain why the idea of “climate debt” can be a useful concept. Under State responsibility, States that have failed to comply with their international obligations (in this case mitigation of emissions) have duties for reparations which can include compensation. I am trying to juxtapose state responsibility as creating a "climate debt" that can be used as a legal justification for sovereign debt relief.

Clarity and Structure- Thank you for this note- I have added additional subheadings particularly in the section discussing the proposed "debt relief for clean ocean energy" section.

Policy Implications- I appreciate your providing this comment. I have added some additional information on how a proposed debt relief program might work including an image of the different steps. I have also added a subheading to discuss the specific limitations to this proposal of which a primary hurdle will be political will.

Case studies- There are no existing debt-for-energy swaps but there are debt-for climate swaps, debt-for-environment swaps, and debt-for health swaps. I have included citations to these and some brief comments from an IMF report that has analyzed Debt-for Climate Swaps.

Conclusion- I have revisited the conclusion to ensure that it reflects major points from the paper. 

Reviewer 3 Report

Pursuing debt swaps requires a detailed analysis of numerous factors, including transaction costs and how they would impact a country's balance of payments, which would differ for each heavily indebted state.

The paper highlights that a "debt swap" to address climate debt is still being explored and would require case-by-case analysis depending on debt swap limits.

Does not comprehensively analyse the potential economic and financial implications of debt-for-clean energy swaps.

Discuss potential challenges or barriers to implementing debt-for-clean energy swaps, such as political or legal obstacles. Overall, while the paper presents the concept of debt-for-clean energy swaps as a potential solution, it does not detail the practicalities and limitations of implementing such swaps.

A literature review is missing.

Statistical tool can be used to represent the data in the table 1,2,3

Need to check grammar and typo mistake 

Author Response

Thank you very much for your helpful comments. I have restructured the section describing the proposal so that it highlights justifications for "debt relief for clean energy" projects, explains how these might work in practice, and  then identifies limitations. The most salient of these limitations is political will but there can also be instances where other mechanisms may be more appropriate than debt relief to achieve the same outcomes such as grants without necessarily the complexity of a debt relief package. I suggest that this does take away some degree of agency from debtor states to participate in investing in their own energy futures.

Thank you for your comment on literature review. I have added observation from the major papers discussing debt-for climate swaps and references.

On the proposal to use more statistical tools rather than tables- I recognize that the article has numerous tables rather than graphs and other representations. 

I wanted to see if the data could speak for itself in so far as illustrating the gaps between the “haves” and “have nots” of renewable energy—most of whom carry unsustainable levels of debt. I wanted readers to be able to see these through the examples of the subset of 22 countries identified as highly indebted poor countries. Based on my research, this information has not yet been compiled in this form before comparing renewable energy generation with carbon intensity. What I am trying to do with  the tables is to give readers the chance to see some of the background trends that are the reason for the larger proposal being made in this paper. The trends that I am trying to highlight are lack of renewable energy among large portion of the global population (Table 1), the low per capita energy use by the poorest countries with a reasonably high level of carbon intensity (Table 2), the heavy financial debt loads of many of the countries that might wish to invest in lower carbon energy (Table 4),  and the apparent slowdown in uptake of renewable energy for many of the poorest countries(Table 5).  While in some respects, these trends are not surprising- the tables provide more concrete data-driven evidence of why a proposal to accelerate clean energy investment is needed particularly for the heavily indebted states who may be understandably tempted to continue investing  in conventional energy development.

Round 2

Reviewer 3 Report

The table can be easily represented using a graphical representation. 

Lots of formatting mistakes 

Missing literature and comparison

Minor editing of English language required

Author Response

Thank you very much for your additional review. I have simplified Table 1. I won't be able to put it into a graphical representation because there are multiple parameters but I have used the Table just to highlight the quantity of renewable energy being produced by a number of mostly high-income states. I have then provided a description of what was previously in Table 1 in the text. The main point of this set of data is to highlight the availability of renewable energy to certain groups of states and the lack of availability in other regions.

As suggested by the the reviewer, I have continued to refine the arguments and discussions of findings. One of the main point of the articles is to focus on the international state responsibility associated with systemic failure to mitigate emissions by many sovereign debt holders and suggest that reparations may be available for energy poor states by assisting these states to make an renewable energy transition through sovereign debt relief.

As also suggested by the author, I have enhanced the section on the literature review and added additional reference reports from credible organizations on the topic of debt relief including the International Monetary Fund.

Thank you for taking the time to provide constructive feedback.

Round 3

Reviewer 3 Report

Discuss the points which may argue that the need for "generous" political will on the part of creditor nations would make these types of debt relief packages a financial non-starter.

Discuss points on “Some states may prefer to deliver conditional climate grants instead of debt relief packages, as they are perceived to be a safer investment in climate projects.”

The debt relief proposal for ocean energy acknowledges that installing wind farms in a handful of coastal countries will not offset pre-existing emissions from financial creditor states, indicating that more effort is needed from "climate debtor" countries to address carbon emissions.

.

The paper does not address the potential challenges or obstacles in implementing a sovereign debt relief package for clean ocean energy.

The paper does not provide a comprehensive analysis of the financial implications or economic feasibility of implementing debt relief for clean ocean energy. 

Major editing in formating and minor in English required 

Author Response

Thank you very much to the external reviewer for comments. Peer review is always an important part of the research process and I am appreciative for the time that the reviewer took to read and comment on how to strengthen this paper. The reviewer suggests discussing the potential challenges in implementing sovereign debt relief package. Thank you for this suggestion. In response to this comment, I have added in section 3.5.3 some additional language indicating that money set aside for debt relief may not be used in clean ocean energy programs and discussing some of the governance challenges associated with any large scale-program like a debt relief program in exchange for domestic clean energy investment. I have also discussed the real possibility that there may not be enough money available  through a debt relief project at one time to achieve an infrastructure outcome. From my perspective, this is likely to be the biggest economic viability hurdle to a debt proposal for clean energy. 

I have clarified the point on political will to suggest that political will is less of a hurdle for the project than some of the other challenges given that creditor states have in the past agreed to a certain amount of debt relief in past multilateral efforts. 

A comprehensive financial/economic analysis is unfortunately beyond the reach of this paper so I have indicated in the paper by noting that there will be financial implications of having a debt relief policy but the specific implications for a given country are beyond the scope of the paper.  I have noted some of the political implications that might arise as a result of different citizen groups having different economic development priorities within a state.

I apologize for the formatting issues. I have been working in the template using Microsoft Word and I have been unable to fix the formatting issues though I have tried.

Many thanks again to the reviewer for their very helpful comments.